# Learning with a Wasserstein Loss

**Charlie Frogner**[*]    **Chiyuan Zhang**[*]
Center for Brains, Minds and Machines
Massachusetts Institute of Technology
frogner@mit.edu, chiyuan@mit.edu

**Hossein Mobahi**
CSAIL
Massachusetts Institute of Technology
hmobahi@csail.mit.edu

**Mauricio Araya-Polo**
Shell International E & P, Inc.
Mauricio.Araya@shell.com

**Tomaso Poggio**
Center for Brains, Minds and Machines
Massachusetts Institute of Technology
tp@ai.mit.edu

## Abstract

Learning to predict multi-label outputs is challenging, but in many problems there is a natural metric on the outputs that can be used to improve predictions. In this paper we develop a loss function for multi-label learning, based on the Wasserstein distance. The Wasserstein distance provides a natural notion of dissimilarity for probability measures. Although optimizing with respect to the exact Wasserstein distance is costly, recent work has described a regularized approximation that is efficiently computed. We describe an efficient learning algorithm based on this regularization, as well as a novel extension of the Wasserstein distance from probability measures to unnormalized measures. We also describe a statistical learning bound for the loss. The Wasserstein loss can encourage smoothness of the predictions with respect to a chosen metric on the output space. We demonstrate this property on a real-data tag prediction problem, using the Yahoo Flickr Creative Commons dataset, outperforming a baseline that doesn't use the metric.

## 1 Introduction

We consider the problem of learning to predict a non-negative measure over a finite set. This problem includes many common machine learning scenarios. In multiclass classification, for example, one often predicts a vector of scores or probabilities for the classes. And in semantic segmentation [1], one can model the segmentation as being the support of a measure defined over the pixel locations. Many problems in which the output of the learning machine is both non-negative and multi-dimensional might be cast as predicting a measure.

We specifically focus on problems in which the output space has a natural metric or similarity structure, which is known (or estimated) *a priori*. In practice, many learning problems have such structure. In the ImageNet Large Scale Visual Recognition Challenge [ILSVRC] [2], for example, the output dimensions correspond to 1000 object categories that have inherent semantic relationships, some of which are captured in the WordNet hierarchy that accompanies the categories. Similarly, in the keyword spotting task from the IARPA Babel speech recognition project, the outputs correspond to keywords that likewise have semantic relationships. In what follows, we will call the similarity structure on the label space the *ground metric* or *semantic similarity*.

Using the ground metric, we can measure prediction performance in a way that is sensitive to relationships between the different output dimensions. For example, confusing dogs with cats might

---

[*]Authors contributed equally.

[1]Code and data are available at http://cbcl.mit.edu/wasserstein.

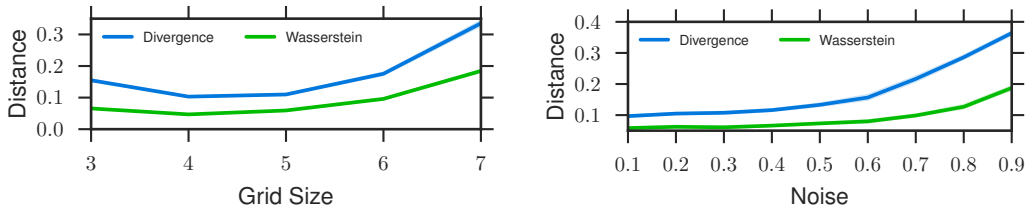

Figure 2: The Wasserstein loss encourages predictions that are similar to ground truth, robustly to incorrect labeling of similar classes (see Appendix E.1). Shown is Euclidean distance between prediction and ground truth vs. (left) number of classes, averaged over different noise levels and (right) noise level, averaged over number of classes. Baseline is the multiclass logistic loss.

be more severe an error than confusing breeds of dogs. A loss function that incorporates this metric might encourage the learning algorithm to favor predictions that are, if not completely accurate, at least semantically similar to the ground truth.

In this paper, we develop a loss function for multi-label learning that measures the *Wasserstein distance* between a prediction and the target label, with respect to a chosen metric on the output space. The Wasserstein distance is defined as the cost of the optimal transport plan for moving the mass in the predicted measure to match that in the target, and has been applied to a wide range of problems, including barycenter estimation [3], label propagation [4], and clustering [5]. To our knowledge, this paper represents the first use of the Wasserstein distance as a loss for supervised learning.

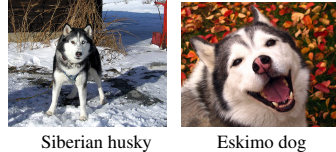

Figure 1: Semantically near-equivalent classes in ILSVRC

We briefly describe a case in which the Wasserstein loss improves learning performance. The setting is a multiclass classification problem in which label noise arises from confusion of semantically near-equivalent categories. Figure 1 shows such a case from the ILSVRC, in which the categories *Siberian husky* and *Eskimo dog* are nearly indistinguishable. We synthesize a toy version of this problem by identifying categories with points in the Euclidean plane and randomly switching the training labels to nearby classes. The Wasserstein loss yields predictions that are closer to the ground truth, robustly across all noise levels, as shown in Figure 2. The standard multiclass logistic loss is the baseline for comparison. Section E.1 in the Appendix describes the experiment in more detail.

The main contributions of this paper are as follows. We formulate the problem of learning with prior knowledge of the ground metric, and propose the Wasserstein loss as an alternative to traditional information divergence-based loss functions. Specifically, we focus on empirical risk minimization (ERM) with the Wasserstein loss, and describe an efficient learning algorithm based on entropic regularization of the optimal transport problem. We also describe a novel extension to unnormalized measures that is similarly efficient to compute. We then justify ERM with the Wasserstein loss by showing a statistical learning bound. Finally, we evaluate the proposed loss on both synthetic examples and a real-world image annotation problem, demonstrating benefits for incorporating an output metric into the loss.

## 2 Related work

Decomposable loss functions like KL Divergence and $\ell_p$ distances are very popular for probabilistic [1] or vector-valued [6] predictions, as each component can be evaluated independently, often leading to simple and efficient algorithms. The idea of exploiting smoothness in the label space according to a prior metric has been explored in many different forms, including regularization [7] and post-processing with graphical models [8]. Optimal transport provides a natural distance for probability distributions over metric spaces. In [3, 9], the optimal transport is used to formulate the Wasserstein barycenter as a probability distribution with minimum total Wasserstein distance to a set of given points on the probability simplex. [4] propagates histogram values on a graph by minimizing a Dirichlet energy induced by optimal transport. The Wasserstein distance is also used to formulate a metric for comparing clusters in [5], and is applied to image retrieval [10], contour

matching [11], and many other problems [12, 13]. However, to our knowledge, this is the first time it is used as a loss function in a discriminative learning framework. The closest work to this paper is a theoretical study [14] of an estimator that minimizes the optimal transport cost between the empirical distribution and the estimated distribution in the setting of statistical parameter estimation.

## 3 Learning with a Wasserstein loss

### 3.1 Problem setup and notation

We consider the problem of learning a map from $\mathcal{X} \subset \mathbb{R}^D$ into the space $\mathcal{Y} = \mathbb{R}_+^K$ of measures over a finite set $\mathcal{K}$ of size $|\mathcal{K}| = K$. Assume $\mathcal{K}$ possesses a metric $d_{\mathcal{K}}(\cdot, \cdot)$, which is called the *ground metric*. $d_{\mathcal{K}}$ measures semantic similarity between dimensions of the output, which correspond to the elements of $\mathcal{K}$. We perform learning over a hypothesis space $\mathcal{H}$ of predictors $h_\theta : \mathcal{X} \to \mathcal{Y}$, parameterized by $\theta \in \Theta$. These might be linear logistic regression models, for example.

In the standard statistical learning setting, we get an i.i.d. sequence of training examples $S = ((x_1, y_1), \ldots, (x_N, y_N))$, sampled from an unknown joint distribution $\mathcal{P}_{\mathcal{X} \times \mathcal{Y}}$. Given a measure of performance (a.k.a. *risk*) $\mathcal{E}(\cdot, \cdot)$, the goal is to find the predictor $h_\theta \in \mathcal{H}$ that minimizes the expected risk $\mathbb{E}[\mathcal{E}(h_\theta(x), y)]$. Typically $\mathcal{E}(\cdot, \cdot)$ is difficult to optimize directly and the joint distribution $\mathcal{P}_{\mathcal{X} \times \mathcal{Y}}$ is unknown, so learning is performed via *empirical risk minimization*. Specifically, we solve

$$\min_{h_\theta \in \mathcal{H}} \left\{ \hat{\mathbb{E}}_S[\ell(h_\theta(x), y) = \frac{1}{N} \sum_{i=1}^{N} \ell(h_\theta(x_i), y_i) \right\} \tag{1}$$

with a loss function $\ell(\cdot, \cdot)$ acting as a surrogate of $\mathcal{E}(\cdot, \cdot)$.

### 3.2 Optimal transport and the exact Wasserstein loss

Information divergence-based loss functions are widely used in learning with probability-valued outputs. Along with other popular measures like Hellinger distance and $\chi^2$ distance, these divergences treat the output dimensions independently, ignoring any metric structure on $\mathcal{K}$.

Given a cost function $c : \mathcal{K} \times \mathcal{K} \to \mathbb{R}$, the *optimal transport* distance [15] measures the cheapest way to transport the mass in probability measure $\mu_1$ to match that in $\mu_2$:

$$W_c(\mu_1, \mu_2) = \inf_{\gamma \in \Pi(\mu_1, \mu_2)} \int_{\mathcal{K} \times \mathcal{K}} c(\kappa_1, \kappa_2) \gamma(d\kappa_1, d\kappa_2) \tag{2}$$

where $\Pi(\mu_1, \mu_2)$ is the set of joint probability measures on $\mathcal{K} \times \mathcal{K}$ having $\mu_1$ and $\mu_2$ as marginals. An important case is that in which the cost is given by a metric $d_{\mathcal{K}}(\cdot, \cdot)$ or its $p$-th power $d_{\mathcal{K}}^p(\cdot, \cdot)$ with $p \geq 1$. In this case, (2) is called a *Wasserstein distance* [16], also known as the *earth mover's distance* [10]. In this paper, we only work with discrete measures. In the case of probability measures, these are histograms in the simplex $\Delta^{\mathcal{K}}$. When the ground truth $y$ and the output of $h$ both lie in the simplex $\Delta^{\mathcal{K}}$, we can define a Wasserstein loss.

**Definition 3.1** (Exact Wasserstein Loss). *For any $h_\theta \in \mathcal{H}$, $h_\theta : \mathcal{X} \to \Delta^{\mathcal{K}}$, let $h_\theta(\kappa|x) = h_\theta(x)_\kappa$ be the predicted value at element $\kappa \in \mathcal{K}$, given input $x \in \mathcal{X}$. Let $y(\kappa)$ be the ground truth value for $\kappa$ given by the corresponding label $y$. Then we define the* exact Wasserstein loss *as*

$$W_p^p(h(\cdot|x), y(\cdot)) = \inf_{T \in \Pi(h(x), y)} \langle T, M \rangle \tag{3}$$

*where $M \in \mathbb{R}_+^{K \times K}$ is the distance matrix $M_{\kappa, \kappa'} = d_{\mathcal{K}}^p(\kappa, \kappa')$, and the set of valid transport plans is*

$$\Pi(h(x), y) = \{T \in \mathbb{R}_+^{K \times K} : T\mathbf{1} = h(x), \ T^\top \mathbf{1} = y\} \tag{4}$$

*where $\mathbf{1}$ is the all-one vector.*

$W_p^p$ is the cost of the optimal plan for transporting the predicted mass distribution $h(x)$ to match the target distribution $y$. The penalty increases as more mass is transported over longer distances, according to the ground metric $M$.

---

**Algorithm 1** Gradient of the Wasserstein loss

---

Given $h(x)$, $y$, $\lambda$, $\mathbf{K}$. ($\gamma_a$, $\gamma_b$ if $h(x)$, $y$ unnormalized.)

$u \leftarrow \mathbf{1}$

**while** $u$ has not converged **do**

$$u \leftarrow \begin{cases} h(x) \oslash \left( \mathbf{K} \left( y \oslash \mathbf{K}^\top u \right) \right) & \text{if } h(x), y \text{ normalized} \\ h(x)^{\frac{\gamma_a \lambda}{\gamma_a \lambda + 1}} \oslash \left( \mathbf{K} \left( y \oslash \mathbf{K}^\top u \right)^{\frac{\gamma_b \lambda}{\gamma_b \lambda + 1}} \right)^{\frac{\gamma_a \lambda}{\gamma_a \lambda + 1}} & \text{if } h(x), y \text{ unnormalized} \end{cases}$$

**end while**

If $h(x)$, $y$ unnormalized: $v \leftarrow y^{\frac{\gamma_b \lambda}{\gamma_b \lambda + 1}} \oslash \left( \mathbf{K}^\top u \right)^{\frac{\gamma_b \lambda}{\gamma_b \lambda + 1}}$

$$\partial W_p^p / \partial h(x) \leftarrow \begin{cases} \frac{\log u}{\lambda} - \frac{\log u^\top \mathbf{1}}{\lambda K} \mathbf{1} & \text{if } h(x), y \text{ normalized} \\ \gamma_a \left( \mathbf{1} - (\text{diag}(u) \mathbf{K} v) \oslash h(x) \right) & \text{if } h(x), y \text{ unnormalized} \end{cases}$$

---

## 4 Efficient optimization via entropic regularization

To do learning, we optimize the empirical risk minimization functional (1) by gradient descent. Doing so requires evaluating a descent direction for the loss, with respect to the predictions $h(x)$. Unfortunately, computing a subgradient of the exact Wasserstein loss (3), is quite costly, as follows.

The exact Wasserstein loss (3) is a linear program and a subgradient of its solution can be computed using Lagrange duality. The dual LP of (3) is

$$^d W_p^p(h(x), y) = \sup_{\alpha, \beta \in C_M} \alpha^\top h(x) + \beta^\top y, \quad C_M = \{(\alpha, \beta) \in \mathbb{R}^{K \times K} : \alpha_\kappa + \beta_{\kappa'} \le M_{\kappa, \kappa'}\}. \quad (5)$$

As (3) is a linear program, at an optimum the values of the dual and the primal are equal (see, e.g. [17]), hence the dual optimal $\alpha$ is a subgradient of the loss with respect to its first argument.

Computing $\alpha$ is costly, as it entails solving a linear program with $O(K^2)$ contraints, with $K$ being the dimension of the output space. This cost can be prohibitive when optimizing by gradient descent.

### 4.1 Entropic regularization of optimal transport

Cuturi [18] proposes a smoothed transport objective that enables efficient approximation of both the transport matrix in (3) and the subgradient of the loss. [18] introduces an entropic regularization term that results in a strictly convex problem:

$$^\lambda W_p^p(h(\cdot|x), y(\cdot)) = \inf_{T \in \Pi(h(x), y)} \langle T, M \rangle - \frac{1}{\lambda} H(T), \quad H(T) = -\sum_{\kappa, \kappa'} T_{\kappa, \kappa'} \log T_{\kappa, \kappa'}. \quad (6)$$

Importantly, the transport matrix that solves (6) is a *diagonal scaling* of a matrix $\mathbf{K} = e^{-\lambda M - 1}$:

$$T^* = \text{diag}(u) \mathbf{K} \text{diag}(v) \quad (7)$$

for $u = e^{\lambda \alpha}$ and $v = e^{\lambda \beta}$, where $\alpha$ and $\beta$ are the Lagrange dual variables for (6).

Identifying such a matrix subject to equality constraints on the row and column sums is exactly a *matrix balancing* problem, which is well-studied in numerical linear algebra and for which efficient iterative algorithms exist [19]. [18] and [3] use the well-known Sinkhorn-Knopp algorithm.

### 4.2 Extending smoothed transport to the learning setting

When the output vectors $h(x)$ and $y$ lie in the simplex, (6) can be used directly in place of (3), as (6) can approximate the exact Wasserstein distance closely for large enough $\lambda$ [18]. In this case, the gradient $\alpha$ of the objective can be obtained from the optimal scaling vector $u$ as $\alpha = \frac{\log u}{\lambda} - \frac{\log u^\top \mathbf{1}}{\lambda K} \mathbf{1}$.
[1] A Sinkhorn iteration for the gradient is given in Algorithm 1.

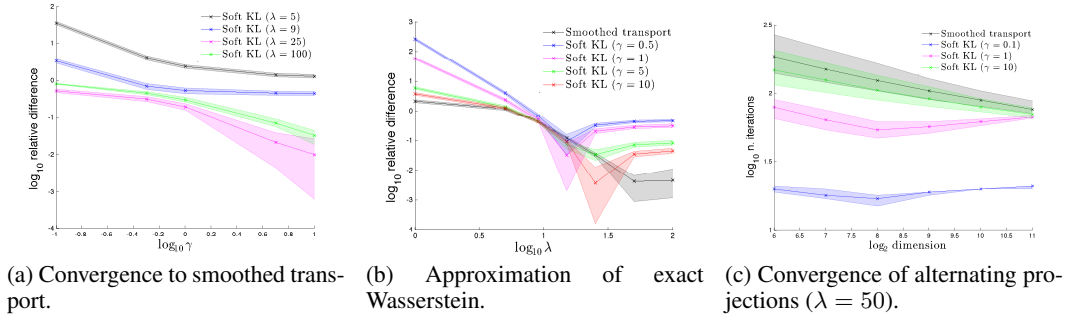

(a) Convergence to smoothed transport.

(b) Approximation of exact Wasserstein.

(c) Convergence of alternating projections ($\lambda = 50$).

Figure 3: The relaxed transport problem (8) for unnormalized measures.

For many learning problems, however, a normalized output assumption is unnatural. In image segmentation, for example, the target shape is not naturally represented as a histogram. And even when the prediction and the ground truth are constrained to the simplex, the observed label can be subject to noise that violates the constraint.

There is more than one way to generalize optimal transport to unnormalized measures, and this is a subject of active study [20]. We will develop here a novel objective that deals effectively with the difference in total mass between $h(x)$ and $y$ while still being efficient to optimize.

### 4.3 Relaxed transport

We propose a novel relaxation that extends smoothed transport to unnormalized measures. By replacing the equality constraints on the transport marginals in (6) with soft penalties with respect to KL divergence, we get an unconstrained approximate transport problem. The resulting objective is:

$$^{\lambda,\gamma_a,\gamma_b}W_{KL}(h(\cdot|x),y(\cdot)) = \min_{T \in \mathbb{R}_+^{K \times K}} \langle T, M \rangle - \frac{1}{\lambda}H(T) + \gamma_a \widetilde{\mathrm{KL}}\left(T\mathbf{1} \| h(x)\right) + \gamma_b \widetilde{\mathrm{KL}}\left(T^\top \mathbf{1} \| y\right) \quad (8)$$

where $\widetilde{\mathrm{KL}}(w \| z) = w^\top \log(w \oslash z) - \mathbf{1}^\top w + \mathbf{1}^\top z$ is the *generalized KL divergence* between $w, z \in \mathbb{R}_+^K$. Here $\oslash$ represents element-wise division. As with the previous formulation, the optimal transport matrix with respect to (8) is a diagonal scaling of the matrix $\mathbf{K}$.

**Proposition 4.1.** *The transport matrix $T^*$ optimizing* (8) *satisfies* $T^* = \mathrm{diag}(u)\mathbf{K}\mathrm{diag}(v)$, *where* $u = (h(x) \oslash T^*\mathbf{1})^{\gamma_a \lambda}$, $v = \left(y \oslash (T^*)^\top \mathbf{1}\right)^{\gamma_b \lambda}$, *and* $\mathbf{K} = e^{-\lambda M - 1}$.

And the optimal transport matrix is a fixed point for a Sinkhorn-like iteration. [2]

**Proposition 4.2.** $T^* = \mathrm{diag}(u)\mathbf{K}\mathrm{diag}(v)$ *optimizing* (8) *satisfies: i)* $u = h(x)^{\frac{\gamma_a \lambda}{\gamma_a \lambda + 1}} \odot (\mathbf{K}v)^{-\frac{\gamma_a \lambda}{\gamma_a \lambda + 1}}$, *and ii)* $v = y^{\frac{\gamma_b \lambda}{\gamma_b \lambda + 1}} \odot \left(\mathbf{K}^\top u\right)^{-\frac{\gamma_b \lambda}{\gamma_b \lambda + 1}}$, *where* $\odot$ *represents element-wise multiplication.*

Unlike the previous formulation, (8) is unconstrained with respect to $h(x)$. The gradient is given by $\nabla_{h(x)} W_{KL}(h(\cdot|x), y(\cdot)) = \gamma_a (\mathbf{1} - T^*\mathbf{1} \oslash h(x))$. The iteration is given in Algorithm 1.

When restricted to normalized measures, the relaxed problem (8) approximates smoothed transport (6). Figure 3a shows, for normalized $h(x)$ and $y$, the relative distance between the values of (8) and (6) [3]. For $\lambda$ large enough, (8) converges to (6) as $\gamma_a$ and $\gamma_b$ increase.

(8) also retains two properties of smoothed transport (6). Figure 3b shows that, for normalized outputs, the relaxed loss converges to the unregularized Wasserstein distance as $\lambda$, $\gamma_a$ and $\gamma_b$ increase [4]. And Figure 3c shows that convergence of the iterations in (4.2) is nearly independent of the dimension $K$ of the output space.

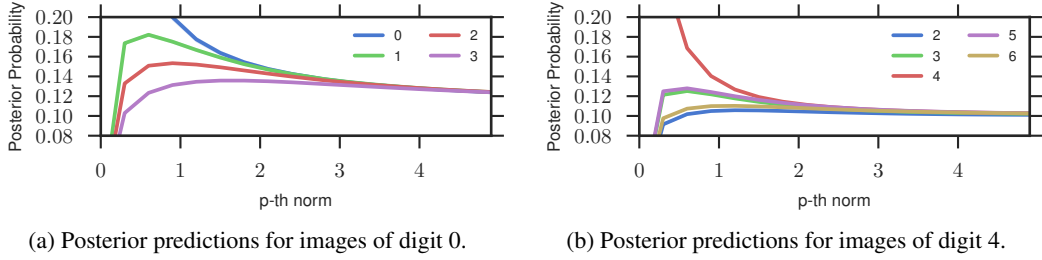

(a) Posterior predictions for images of digit 0.          (b) Posterior predictions for images of digit 4.

Figure 4: MNIST example. Each curve shows the predicted probability for one digit, for models trained with different $p$ values for the ground metric.

# 5  Statistical Properties of the Wasserstein loss

Let $S = ((x_1, y_1), \ldots, (x_N, y_N))$ be i.i.d. samples and $h_{\hat{\theta}}$ be the empirical risk minimizer

$$h_{\hat{\theta}} = \underset{h_\theta \in \mathcal{H}}{\mathrm{argmin}} \left\{ \hat{\mathbb{E}}_S \left[ W_p^p(h_\theta(\cdot|x), y) \right] = \frac{1}{N} \sum_{i=1}^{N} W_p^p(h_x \theta(\cdot|x_i), y_i) \right\}.$$

Further assume $\mathcal{H} = \mathfrak{s} \circ \mathcal{H}^o$ is the composition of a softmax $\mathfrak{s}$ and a base hypothesis space $\mathcal{H}^o$ of functions mapping into $\mathbb{R}^K$. The softmax layer outputs a prediction that lies in the simplex $\Delta^{\mathcal{K}}$.

**Theorem 5.1.** *For $p = 1$, and any $\delta > 0$, with probability at least $1 - \delta$, it holds that*

$$\mathbb{E} \left[ W_1^1(h_{\hat{\theta}}(\cdot|x), y) \right] \leq \inf_{h_\theta \in \mathcal{H}} \mathbb{E} \left[ W_1^1(h_\theta(\cdot|x), y) \right] + 32KC_M \mathfrak{R}_N(\mathcal{H}^o) + 2C_M \sqrt{\frac{\log(1/\delta)}{2N}} \quad (9)$$

*with the constant $C_M = \max_{\kappa, \kappa'} M_{\kappa, \kappa'}$. $\mathfrak{R}_N(\mathcal{H}^o)$ is the* Rademacher *complexity [22] measuring the complexity of the hypothesis space $\mathcal{H}^o$.*

The Rademacher complexity $\mathfrak{R}_N(\mathcal{H}^o)$ for commonly used models like neural networks and kernel machines [22] decays with the training set size. This theorem guarantees that the expected Wasserstein loss of the empirical risk minimizer approaches the best achievable loss for $\mathcal{H}$.

As an important special case, minimizing the empirical risk with Wasserstein loss is also good for multiclass classification. Let $y = \mathbb{e}_\kappa$ be the "one-hot" encoded label vector for the groundtruth class.

**Proposition 5.2.** *In the multiclass classification setting, for $p = 1$ and any $\delta > 0$, with probability at least $1 - \delta$, it holds that*

$$\mathbb{E}_{x,\kappa} \left[ d_\mathcal{K}(\kappa_{\hat{\theta}}(x), \kappa) \right] \leq \inf_{h_\theta \in \mathcal{H}} K\mathbb{E}[W_1^1(h_\theta(x), y)] + 32K^2 C_M \mathfrak{R}_N(\mathcal{H}^o) + 2C_M K \sqrt{\frac{\log(1/\delta)}{2N}} \quad (10)$$

*where the predictor is $\kappa_{\hat{\theta}}(x) = \mathrm{argmax}_\kappa h_{\hat{\theta}}(\kappa|x)$, with $h_{\hat{\theta}}$ being the empirical risk minimizer.*

Note that instead of the classification error $\mathbb{E}_{x,\kappa}[\mathbb{1}\{\kappa_{\hat{\theta}}(x) \neq \kappa\}]$, we actually get a bound on the expected semantic distance between the prediction and the groundtruth.

# 6  Empirical study

## 6.1  Impact of the ground metric

In this section, we show that the Wasserstein loss encourages smoothness with respect to an artificial metric on the MNIST handwritten digit dataset. This is a multi-class classification problem with output dimensions corresponding to the 10 digits, and we apply a ground metric $d_p(\kappa, \kappa') = |\kappa - \kappa'|^p$, where $\kappa, \kappa' \in \{0, \ldots, 9\}$ and $p \in [0, \infty)$. This metric encourages the recognized digit to be *numerically* close to the true one. We train a model independently for each value of $p$ and plot the average predicted probabilities of the different digits on the test set in Figure 4.

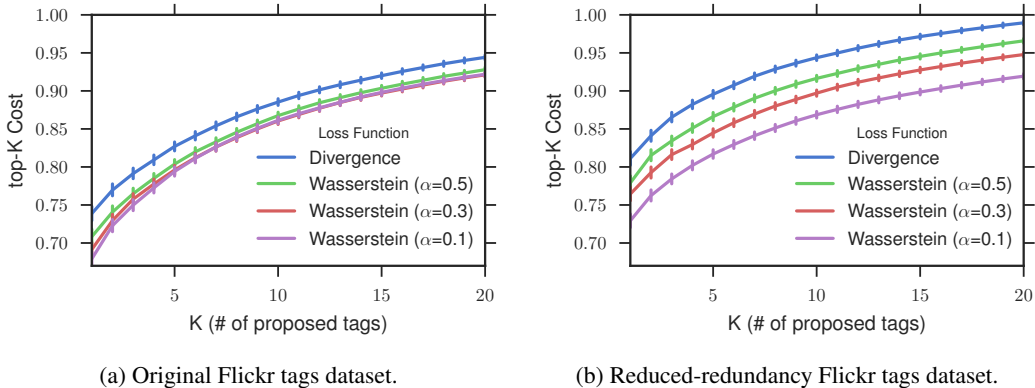

(a) Original Flickr tags dataset.

(b) Reduced-redundancy Flickr tags dataset.

Figure 5: Top-K cost comparison of the proposed loss (Wasserstein) and the baseline (Divergence).

Note that as $p \to 0$, the metric approaches the $0 - 1$ metric $d_0(\kappa, \kappa') = \mathbb{1}_{\kappa \neq \kappa'}$, which treats all incorrect digits as being equally unfavorable. In this case, as can be seen in the figure, the predicted probability of the true digit goes to 1 while the probability for all other digits goes to 0. As $p$ increases, the predictions become more evenly distributed over the neighboring digits, converging to a uniform distribution as $p \to \infty$ [5].

## 6.2 Flickr tag prediction

We apply the Wasserstein loss to a real world multi-label learning problem, using the recently released Yahoo/Flickr Creative Commons 100M dataset [23]. [6] Our goal is *tag prediction*: we select 1000 descriptive tags along with two random sets of 10,000 images each, associated with these tags, for training and testing. We derive a distance metric between tags by using `word2vec` [24] to embed the tags as unit vectors, then taking their Euclidean distances. To extract image features we use `MatConvNet` [25]. Note that the set of tags is highly redundant and often many semantically equivalent or similar tags can apply to an image. The images are also partially tagged, as different users may prefer different tags. We therefore measure the prediction performance by the *top-K cost*, defined as $C_K = 1/K \sum_{k=1}^{K} \min_j d_{\mathcal{K}}(\hat{\kappa}_k, \kappa_j)$, where $\{\kappa_j\}$ is the set of groundtruth tags, and $\{\hat{\kappa}_k\}$ are the tags with highest predicted probability. The standard AUC measure is also reported.

We find that a linear combination of the Wasserstein loss $W_p^p$ and the standard multiclass logistic loss KL yields the best prediction results. Specifically, we train a linear model by minimizing $W_p^p + \alpha$KL on the training set, where $\alpha$ controls the relative weight of KL. Note that KL taken alone is our baseline in these experiments. Figure 5a shows the top-K cost on the test set for the combined loss and the baseline KL loss. We additionally create a second dataset by removing redundant labels from the original dataset: this simulates the potentially more difficult case in which a single user tags each image, by selecting one tag to apply from amongst each cluster of applicable, semantically similar tags. Figure 3b shows that performance for both algorithms decreases on the harder dataset, while the combined Wasserstein loss continues to outperform the baseline.

In Figure 6, we show the effect on performance of varying the weight $\alpha$ on the KL loss. We observe that the optimum of the top-$K$ cost is achieved when the Wasserstein loss is weighted more heavily than at the optimum of the AUC. This is consistent with a semantic smoothing effect of Wasserstein, which during training will favor mispredictions that are semantically similar to the ground truth, sometimes at the cost of lower AUC [7]. We finally show two selected images from the test set in Figure 7. These illustrate cases in which both algorithms make predictions that are semantically relevant, despite overlapping very little with the ground truth. The image on the left shows errors made by both algorithms. More examples can be found in the appendix.

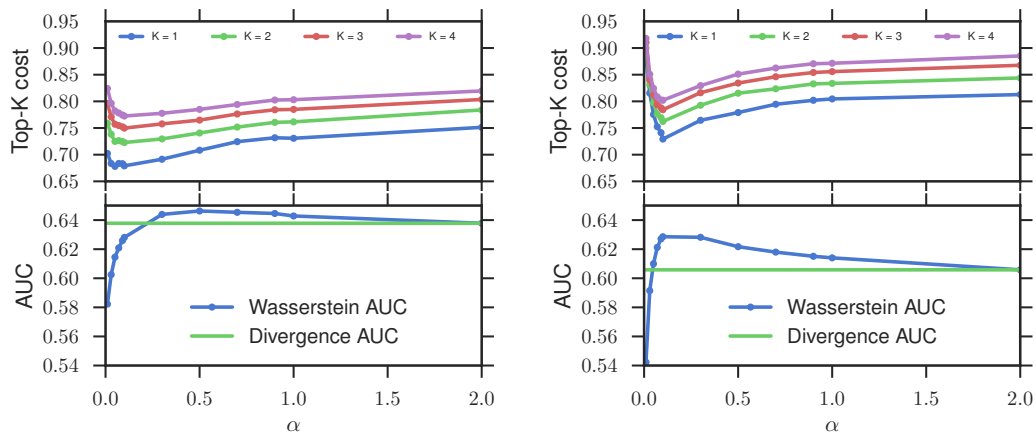

(a) Original Flickr tags dataset.　　　　(b) Reduced-redundancy Flickr tags dataset.

Figure 6: Trade-off between semantic smoothness and maximum likelihood.

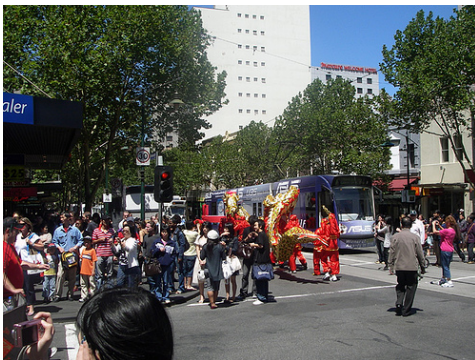

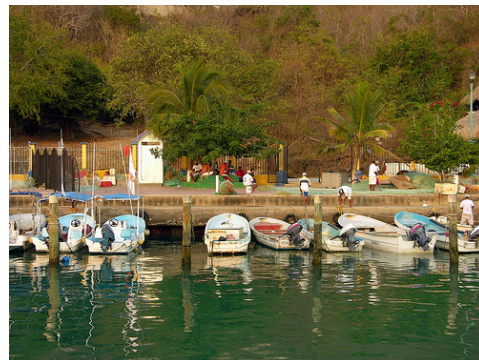

(a) **Flickr user tags**: street, parade, dragon; **our proposals**: people, protest, parade; **baseline proposals**: music, car, band.

(b) **Flickr user tags**: water, boat, reflection, sunshine; **our proposals**: water, river, lake, summer; **baseline proposals**: river, water, club, nature.

Figure 7: Examples of images in the Flickr dataset. We show the groundtruth tags and as well as tags proposed by our algorithm and the baseline.

## 7    Conclusions and future work

In this paper we have described a loss function for learning to predict a non-negative measure over a finite set, based on the Wasserstein distance. Although optimizing with respect to the exact Wasserstein loss is computationally costly, an approximation based on entropic regularization is efficiently computed. We described a learning algorithm based on this regularization and we proposed a novel extension of the regularized loss to unnormalized measures that preserves its efficiency. We also described a statistical learning bound for the loss. The Wasserstein loss can encourage smoothness of the predictions with respect to a chosen metric on the output space, and we demonstrated this property on a real-data tag prediction problem, showing improved performance over a baseline that doesn't incorporate the metric.

An interesting direction for future work may be to explore the connection between the Wasserstein loss and Markov random fields, as the latter are often used to encourage smoothness of predictions, via inference at prediction time.

## Footnotes

[1]Note that $\alpha$ is only defined up to a constant shift: any upscaling of the vector $u$ can be paired with a corresponding downscaling of the vector $v$ (and vice versa) without altering the matrix $T^*$. The choice $\alpha = \frac{\log u}{\lambda} - \frac{\log u^\top \mathbf{1}}{\lambda K} \mathbf{1}$ ensures that $\alpha$ is tangent to the simplex.

[2]Note that, although the iteration suggested by Proposition 4.2 is observed empirically to converge (see Figure 3c, for example), we have not proven a guarantee that it will do so.

[3]In figures 3a-c, $h(x)$, $y$ and $M$ are generated as described in [18] section 5. In 3a-b, $h(x)$ and $y$ have dimension 256. In 3c, convergence is defined as in [18]. Shaded regions are 95% intervals.

[4]The unregularized Wasserstein distance was computed using `FastEMD` [21].

[5]To avoid numerical issues, we scale down the ground metric such that all of the distance values are in the interval $[0, 1)$.

[6]The dataset used here is available at `http://cbcl.mit.edu/wasserstein`.

[7]The Wasserstein loss can achieve a similar trade-off by choosing the metric parameter $p$, as discussed in Section 6.1. However, the relationship between $p$ and the smoothing behavior is complex and it can be simpler to implement the trade-off by combining with the KL loss.

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
