[Supplementary Material]

# A Relaxed transport

Equation (8) gives the relaxed transport objective as

$$^{\lambda,\gamma_a,\gamma_b}W_{KL}(h(\cdot|x),y(\cdot)) = \min_{T\in\mathbb{R}_+^{K\times K}}\langle T, M\rangle - \frac{1}{\lambda}H(T) + \gamma_a\widetilde{\text{KL}}\left(T\mathbf{1}\|h(x)\right) + \gamma_b\widetilde{\text{KL}}\left(T^\top\mathbf{1}\|y\right)$$

with $\widetilde{\text{KL}}\left(w\|z\right) = w^\top\log(w\oslash z) - \mathbf{1}^\top w + \mathbf{1}^\top z$.

*Proof of Proposition 4.1.* The first order condition for $T^*$ optimizing (8) is

$$M_{ij} + \frac{1}{\lambda}\left(\log T_{ij}^* + 1\right) + \gamma_a\left(\log T^*\mathbf{1}\oslash h(x)\right)_i + \gamma_b\left(\log(T^*)^\top\mathbf{1}\oslash y\right)_j = 0.$$

$$\Rightarrow \log T_{ij}^* + \gamma_a\lambda\log\left(T^*\mathbf{1}\oslash h(x_i)\right)_i + \gamma_b\lambda\log\left((T^*)^\top\mathbf{1}\oslash y_j\right)_j = -\lambda M_{ij} - 1$$

$$\Rightarrow T_{ij}^*\left(T^*\mathbf{1}\oslash h(x)\right)_i^{\gamma_a\lambda}\left((T^*)^\top\mathbf{1}\oslash y\right)_j^{\gamma_b\lambda} = \exp\left(-\lambda M_{ij} - 1\right)$$

$$\Rightarrow T_{ij}^* = \left(h(x)\oslash T^*\mathbf{1}\right)_i^{\gamma_a\lambda}\left(y\oslash(T^*)^\top\mathbf{1}\right)_j^{\gamma_b\lambda}\exp\left(-\lambda M_{ij} - 1\right)$$

Hence $T^*$ (if it exists) is a diagonal scaling of $\mathbf{K} = \exp\left(-\lambda M - 1\right)$.

$\square$

*Proof of Proposition 4.2.* Let $u = \left(h(x)\oslash T^*\mathbf{1}\right)^{\gamma_a\lambda}$ and $v = \left(y\oslash(T^*)^\top\mathbf{1}\right)^{\gamma_b\lambda}$, so $T^* = \text{diag}(u)\mathbf{K}\text{diag}(v)$. We have

$$T^*\mathbf{1} = \text{diag}(u)\mathbf{K}v$$

$$\Rightarrow (T^*\mathbf{1})^{\gamma_a\lambda+1} = h(x)^{\gamma_a\lambda}\odot\mathbf{K}v$$

where we substituted the expression for $u$. Re-writing $T^*\mathbf{1}$,

$$(\text{diag}(u)\mathbf{K}v)^{\gamma_a\lambda+1} = \text{diag}(h(x)^{\gamma_a\lambda})\mathbf{K}v$$

$$\Rightarrow u^{\gamma_a\lambda+1} = h(x)^{\gamma_a\lambda}\odot(\mathbf{K}v)^{-\gamma_a\lambda}$$

$$\Rightarrow u = h(x)^{\frac{\gamma_a\lambda}{\gamma_a\lambda+1}}\odot(\mathbf{K}v)^{-\frac{\gamma_a\lambda}{\gamma_a\lambda+1}}.$$

A symmetric argument shows that $v = y^{\frac{\gamma_b\lambda}{\gamma_b\lambda+1}}\odot(\mathbf{K}^\top u)^{-\frac{\gamma_b\lambda}{\gamma_b\lambda+1}}$. $\square$

# B Statistical Learning Bounds

We establish the proof of Theorem 5.1 in this section. For simpler notation, for a sequence $S = ((x_1,y_1),\ldots,(x_N,y_N))$ of i.i.d. training samples, we denote the empirical risk $\hat{R}_S$ and risk $R$ as

$$\hat{R}_S(h_\theta) = \hat{\mathbb{E}}_S\left[W_p^p(h_\theta(\cdot|x),y(\cdot))\right], \quad R(h_\theta) = \mathbb{E}\left[W_p^p(h_\theta(\cdot|x),y(\cdot))\right] \qquad (11)$$

**Lemma B.1.** *Let $h_{\hat{\theta}}, h_{\theta^*}\in\mathcal{H}$ be the minimizer of the empirical risk $\hat{R}_S$ and expected risk $R$, respectively. Then*

$$R(h_{\hat{\theta}}) \leq R(h_{\theta^*}) + 2\sup_{h\in\mathcal{H}}|R(h) - \hat{R}_S(h)|$$

*Proof.* By the optimality of $h_{\hat{\theta}}$ for $\hat{R}_S$,

$$R(h_{\hat{\theta}}) - R(h_{\theta^*}) = R(h_{\hat{\theta}}) - \hat{R}_S(h_{\hat{\theta}}) + \hat{R}_S(h_{\hat{\theta}}) - R(h_{\theta^*})$$

$$\leq R(h_{\hat{\theta}}) - \hat{R}_S(h_{\hat{\theta}}) + \hat{R}_S(h_{\theta^*}) - R(h_{\theta^*})$$

$$\leq 2\sup_{h\in\mathcal{H}}|R(h) - \hat{R}_S(h)|$$

$\square$

Therefore, to bound the risk for $h_{\hat{\theta}}$, we need to establish uniform concentration bounds for the Wasserstein loss. Towards that goal, we define a space of loss functions induced by the hypothesis space $\mathcal{H}$ as

$$\mathcal{L} = \left\{ \ell_\theta : (x, y) \mapsto W_p^p(h_\theta(\cdot|x), y(\cdot)) : h_\theta \in \mathcal{H} \right\} \tag{12}$$

The uniform concentration will depends on the "complexity" of $\mathcal{L}$, which is measured by the empirical *Rademacher complexity* defined below.

**Definition B.2** (Rademacher Complexity [22]). *Let $\mathcal{G}$ be a family of mapping from $\mathcal{Z}$ to $\mathbb{R}$, and $S = (z_1, \ldots, z_N)$ a fixed sample from $\mathcal{Z}$. The* empirical Rademacher complexity *of $\mathcal{G}$ with respect to $S$ is defined as*

$$\hat{\mathfrak{R}}_S(\mathcal{G}) = \mathbb{E}_\sigma \left[ \sup_{g \in \mathcal{G}} \frac{1}{N} \sum_{i=1}^{n} \sigma_i g(z_i) \right] \tag{13}$$

*where $\sigma = (\sigma_1, \ldots, \sigma_N)$, with $\sigma_i$'s independent uniform random variables taking values in $\{+1, -1\}$. $\sigma_i$'s are called the Rademacher random variables. The* Rademacher complexity *is defined by taking expectation with respect to the samples $S$,*

$$\mathfrak{R}_N(\mathcal{G}) = \mathbb{E}_S \left[ \hat{\mathfrak{R}}_S(\mathcal{G}) \right] \tag{14}$$

**Theorem B.3.** *For any $\delta > 0$, with probability at least $1 - \delta$, the following holds for all $\ell_\theta \in \mathcal{L}$,*

$$\mathbb{E}[\ell_\theta] - \hat{\mathbb{E}}_S[\ell_\theta] \leq 2\mathfrak{R}_N(\mathcal{L}) + \sqrt{\frac{C_M^2 \log(1/\delta)}{2N}} \tag{15}$$

*with the constant $C_M = \max_{\kappa, \kappa'} M_{\kappa, \kappa'}$.*

By the definition of $\mathcal{L}$, $\mathbb{E}[\ell_\theta] = R(h_\theta)$ and $\hat{\mathbb{E}}_S[\ell_\theta] = \hat{R}_S[h_\theta]$. Therefore, this theorem provides a uniform control for the deviation of the empirical risk from the risk.

**Theorem B.4** (McDiarmid's Inequality). *Let $S = \{X_1, \ldots, X_N\} \subset \mathcal{X}$ be $N$ i.i.d. random variables. Assume there exists $C > 0$ such that $f : \mathcal{X}^N \to \mathbb{R}$ satisfies the following stability condition*

$$|f(x_1, \ldots, x_i, \ldots, x_N) - f(x_1, \ldots, x_i', \ldots, x_N)| \leq C \tag{16}$$

*for all $i = 1, \ldots, N$ and any $x_1, \ldots, x_N, x_i' \in \mathcal{X}$. Then for any $\varepsilon > 0$, denoting $f(X_1, \ldots, X_N)$ by $f(S)$, it holds that*

$$\mathbb{P}\left( f(S) - \mathbb{E}[f(S)] \geq \varepsilon \right) \leq \exp\left( -\frac{2\varepsilon^2}{NC^2} \right) \tag{17}$$

**Lemma B.5.** *Let the constant $C_M = \max_{\kappa, \kappa'} M_{\kappa, \kappa'}$, then $0 \leq W_p^p(\cdot, \cdot) \leq C_M$.*

*Proof.* For any $h(\cdot|x)$ and $y(\cdot)$, let $T^* \in \Pi(h(x), y)$ be the optimal transport plan that solves (3), then

$$W_p^p(h(x), y) = \langle T^*, M \rangle \leq C_M \sum_{\kappa, \kappa'} T_{\kappa, \kappa'} = C_M$$

$\square$

*Proof of Theorem B.3.* For any $\ell_\theta \in \mathcal{L}$, note the empirical expectation is the empirical risk of the corresponding $h_\theta$:

$$\hat{E}_S[\ell_\theta] = \frac{1}{N} \sum_{i=1}^{N} \ell_\theta(x_i, y_i) = \frac{1}{N} \sum_{i=1}^{N} W_p^p(h_\theta(\cdot|x_i), y_i(\cdot)) = \hat{R}_S(h_\theta)$$

Similarly, $\mathbb{E}[\ell_\theta] = R(h_\theta)$. Let

$$\Phi(S) = \sup_{\ell \in \mathcal{L}} \mathbb{E}[\ell] - \hat{\mathbb{E}}_S[\ell] \tag{18}$$

Let $S'$ be $S$ with the $i$-th sample replaced by $(x_i', y_i')$, by Lemma B.5, it holds that

$$\Phi(S) - \Phi(S') \leq \sup_{\ell \in \mathcal{L}} \hat{\mathbb{E}}_{S'}[\ell] - \hat{\mathbb{E}}_S[\ell] = \sup_{h_\theta \in \mathcal{H}} \frac{W_p^p(h_\theta(x_i'), y_i') - W_p^p(h_\theta(x_i), y_i)}{N} \leq \frac{C_M}{N}$$

Similarly, we can show $\Phi(S') - \Phi(S) \leq C_M/N$, thus $|\Phi(S') - \Phi(S)| \leq C_M/N$. By Theorem B.4, for any $\delta > 0$, with probability at least $1 - \delta$, it holds that

$$\Phi(S) \leq \mathbb{E}[\Phi(S)] + \sqrt{\frac{C_M^2 \log(1/\delta)}{2N}} \tag{19}$$

To bound $\mathbb{E}[\Phi(S)]$, by Jensen's inequality,

$$\mathbb{E}_S[\Phi(S)] = \mathbb{E}_S \left[ \sup_{\ell \in \mathcal{L}} \mathbb{E}[\ell] - \hat{E}_S[\ell] \right] = \mathbb{E}_S \left[ \sup_{\ell \in \mathcal{L}} \mathbb{E}_{S'} \left[ \hat{\mathbb{E}}_{S'}[\ell] - \hat{E}_S[\ell] \right] \right] \leq \mathbb{E}_{S,S'} \left[ \sup_{\ell \in \mathcal{L}} \hat{E}_{S'}[\ell] - \hat{E}_S[\ell] \right]$$

Here $S'$ is another sequence of i.i.d. samples, usually called *ghost samples*, that is only used for analysis. Now we introduce the Rademacher variables $\sigma_i$, since the role of $S$ and $S'$ are completely symmetric, it follows

$$\mathbb{E}_S[\Phi(S)] \leq \mathbb{E}_{S,S',\sigma} \left[ \sup_{\ell \in \mathcal{L}} \frac{1}{N} \sum_{i=1}^{N} \sigma_i(\ell(x_i', y_i') - \ell(x_i, y_i)) \right]$$

$$\leq \mathbb{E}_{S',\sigma} \left[ \sup_{\ell \in \mathcal{L}} \frac{1}{N} \sum_{i=1}^{N} \sigma_i \ell(x_i', y_i') \right] + \mathbb{E}_{S,\sigma} \left[ \sup_{\ell \in \mathcal{L}} \frac{1}{N} \sum_{i=1}^{N} -\sigma_i \ell(x_i, y_i) \right]$$

$$= \mathbb{E}_S \left[ \hat{\mathfrak{R}}_S(\mathcal{L}) \right] + \mathbb{E}_{S'} \left[ \hat{\mathfrak{R}}_{S'}(\mathcal{L}) \right]$$

$$= 2\mathfrak{R}_N(\mathcal{L})$$

The conclusion follows by combing (18) and (19). □

To finish the proof of Theorem 5.1, we combine Lemma B.1 and Theorem B.3, and relate $\mathfrak{R}_N(\mathcal{L})$ to $\mathfrak{R}_N(\mathcal{H})$ via the following generalized Talagrand's lemma [26].

**Lemma B.6.** *Let $\mathcal{F}$ be a class of real functions, and $\mathcal{H} \subset \mathcal{F} = \mathcal{F}_1 \times \ldots \times \mathcal{F}_K$ be a $K$-valued function class. If $\mathfrak{m} : \mathbb{R}^K \rightarrow \mathbb{R}$ is a $L_{\mathfrak{m}}$-Lipschitz function and $\mathfrak{m}(0) = 0$, then $\hat{\mathfrak{R}}_S(\mathfrak{m} \circ \mathcal{H}) \leq 2L_{\mathfrak{m}} \sum_{k=1}^{K} \hat{\mathfrak{R}}_S(\mathcal{F}_k)$.*

**Theorem B.7** (Theorem 6.15 of [15]). *Let $\mu$ and $\nu$ be two probability measures on a Polish space $(\mathcal{K}, d_\mathcal{K})$. Let $p \in [1, \infty)$ and $\kappa_0 \in \mathcal{K}$. Then*

$$W_p(\mu, \nu) \leq 2^{1/p'} \left( \int_\mathcal{K} d_\mathcal{K}(\kappa_0, \kappa) d|\mu - \nu|(\kappa) \right)^{1/p}, \quad \frac{1}{p} + \frac{1}{p'} = 1 \tag{20}$$

**Corollary B.8.** *The Wasserstein loss is Lipschitz continuous in the sense that for any $h_\theta \in \mathcal{H}$, and any $(x, y) \in \mathcal{X} \times \mathcal{Y}$,*

$$W_p^p(h_\theta(\cdot|x), y) \leq 2^{p-1} C_M \sum_{\kappa \in \mathcal{K}} |h_\theta(\kappa|x) - y(\kappa)| \tag{21}$$

*In particular, when $p = 1$, we have*

$$W_1^1(h_\theta(\cdot|x), y) \leq C_M \sum_{\kappa \in \mathcal{K}} |h_\theta(\kappa|x) - y(\kappa)| \tag{22}$$

We cannot apply Lemma B.6 directly to the Wasserstein loss class, because the Wasserstein loss is only defined on probability distributions, so 0 is not a valid input. To get around this problem, we assume the hypothesis space $\mathcal{H}$ used in learning is of the form

$$\mathcal{H} = \{\mathfrak{s} \circ h^o : h^o \in \mathcal{H}^o\} \tag{23}$$

where $\mathcal{H}^o$ is a function class that maps into $\mathbb{R}^K$, and $\mathfrak{s}$ is the softmax function defined as $\mathfrak{s}(o) = (\mathfrak{s}_1(o), \ldots, \mathfrak{s}_K(o))$, with

$$\mathfrak{s}_k(o) = \frac{e^{o_k}}{\sum_j e^{o_j}}, \quad k = 1, \ldots, K \tag{24}$$

The softmax layer produce a valid probability distribution from arbitrary input, and this is consistent with commonly used models such as Logistic Regression and Neural Networks. By working with the $\log$ of the groundtruth labels, we can also add a softmax layer to the labels.

**Lemma B.9** (Proposition 2 of [27])**.** *The Wasserstein distances $W_p(\cdot, \cdot)$ are metrics on the space of probability distributions of $\mathcal{K}$, for all $1 \leq p \leq \infty$.*

**Proposition B.10.** *The map $\iota : \mathbb{R}^K \times \mathbb{R}^K \to \mathbb{R}$ defined by $\iota(y, y') = W_1^1(\mathfrak{s}(y), \mathfrak{s}(y'))$ satisfies*

$$|\iota(y, y') - \iota(\bar{y}, \bar{y}')| \leq 4C_M \|(y, y') - (\bar{y}, \bar{y}')\|_2 \tag{25}$$

*for any $(y, y'), (\bar{y}, \bar{y}') \in \mathbb{R}^K \times \mathbb{R}^K$. And $\iota(0, 0) = 0$.*

*Proof.* For any $(y, y'), (\bar{y}, \bar{y}') \in \mathbb{R}^K \times \mathbb{R}^K$, by Lemma B.9, we can use triangle inequality on the Wasserstein loss,

$$|\iota(y, y') - \iota(\bar{y}, \bar{y}')| = |\iota(y, y') - \iota(\bar{y}, y') + \iota(\bar{y}, y') - \iota(\bar{y}, \bar{y}')| \leq \iota(y, \bar{y}) + \iota(y', \bar{y}')$$

Following Corollary B.8, it continues as

$$|\iota(y, y') - \iota(\bar{y}, \bar{y}')| \leq C_M \left( \|\mathfrak{s}(y) - \mathfrak{s}(\bar{y})\|_1 + \|\mathfrak{s}(y') - \mathfrak{s}(\bar{y}')\|_1 \right) \tag{26}$$

Note for each $k = 1, \ldots, K$, the gradient $\nabla_y \mathfrak{s}_k$ satisfies

$$\|\nabla_y \mathfrak{s}_k\|_2 = \left\| \left( \frac{\partial \mathfrak{s}_k}{\partial y_j} \right)_{j=1}^K \right\|_2 = \left\| (\delta_{kj} \mathfrak{s}_k - \mathfrak{s}_k \mathfrak{s}_j)_{j=1}^K \right\|_2 = \sqrt{\mathfrak{s}_k^2 \sum_{j=1}^K \mathfrak{s}_j^2 + \mathfrak{s}_k^2 (1 - 2\mathfrak{s}_k)} \tag{27}$$

By mean value theorem, $\exists \alpha \in [0, 1]$, such that for $y_\theta = \alpha y + (1 - \alpha) \bar{y}$, it holds that

$$\|\mathfrak{s}(y) - \mathfrak{s}(\bar{y})\|_1 = \sum_{k=1}^K \left| \langle \nabla_y \mathfrak{s}_k |_{y=y_{\alpha_k}}, y - \bar{y} \rangle \right| \leq \sum_{k=1}^K \|\nabla_y \mathfrak{s}_k |_{y=y_{\alpha_k}}\|_2 \|y - \bar{y}\|_2 \leq 2\|y - \bar{y}\|_2$$

because by (27), and the fact that $\sqrt{\sum_j \mathfrak{s}_j^2} \leq \sum_j \mathfrak{s}_j = 1$ and $\sqrt{a + b} \leq \sqrt{a} + \sqrt{b}$ for $a, b \geq 0$, it holds

$$\sum_{k=1}^K \|\nabla_y \mathfrak{s}_k\|_2 = \sum_{k:\mathfrak{s}_k \leq 1/2} \|\nabla_y \mathfrak{s}_k\|_2 + \sum_{k:\mathfrak{s}_k > 1/2} \|\nabla_y \mathfrak{s}_k\|_2$$

$$\leq \sum_{k:\mathfrak{s}_k \leq 1/2} \left( \mathfrak{s}_k + \mathfrak{s}_k \sqrt{1 - 2\mathfrak{s}_k} \right) + \sum_{k:\mathfrak{s}_k > 1/2} \mathfrak{s}_k \leq \sum_{k=1}^K 2\mathfrak{s}_k = 2$$

Similarly, we have $\|\mathfrak{s}(y') - \mathfrak{s}(\bar{y}')\|_1 \leq 2\|y' - \bar{y}'\|_2$, so from (26), we know

$$|\iota(y, y') - \iota(\bar{y}, \bar{y}')| \leq 2C_M(\|y - \bar{y}\|_2 + \|y' - \bar{y}'\|_2) \leq 2\sqrt{2}C_M \left( \|y - \bar{y}\|_2^2 + \|y' - \bar{y}'\|_2^2 \right)^{1/2}$$

then (25) follows immediately. The second conclusion follows trivially as $\mathfrak{s}$ maps the zero vector to a uniform distribution. $\qquad\square$

*Proof of Theorem 5.1.* Consider the loss function space preceded with a softmax layer

$$\mathcal{L} = \{ \iota_\theta : (x, y) \mapsto W_1^1(\mathfrak{s}(h_\theta^o(x)), \mathfrak{s}(y)) : h_\theta^o \in \mathcal{H}^o \}$$

We apply Lemma B.6 to the $4C_M$-Lipschitz continuous function $\iota$ in Proposition B.10 and the function space

$$\underbrace{\mathcal{H}^o \times \ldots \times \mathcal{H}^o}_{K \text{ copies}} \times \underbrace{\mathcal{I} \times \ldots \times \mathcal{I}}_{K \text{ copies}}$$

with $\mathcal{I}$ a singleton function space with only the identity map. It holds

$$\hat{\mathfrak{R}}_S(\mathcal{L}) \leq 8C_M \left( K\hat{\mathfrak{R}}_S(\mathcal{H}^o) + K\hat{\mathfrak{R}}_S(\mathcal{I}) \right) = 8KC_M \hat{\mathfrak{R}}_S(\mathcal{H}^o) \tag{28}$$

because for the identity map, and a sample $S = (y_1, \ldots, y_N)$, we can calculate

$$\hat{\mathfrak{R}}_S(\mathcal{I}) = \mathbb{E}_\sigma \left[ \sup_{f \in \mathcal{I}} \frac{1}{N} \sum_{i=1}^N \sigma_i f(y_i) \right] = \mathbb{E}_\sigma \left[ \frac{1}{N} \sum_{i=1}^N \sigma_i y_i \right] = 0$$

The conclusion of the theorem follows by combining (28) with Theorem B.3 and Lemma B.1. $\qquad\square$

## C Connection with multiclass classification

*Proof of Proposition 5.2.* Given that the label is a "one-hot" vector $y = \mathbb{e}_\kappa$, the set of transport plans (4) degenerates. Specifically, the constraint $T^\top \mathbf{1} = \mathbb{e}_\kappa$ means that only the $\kappa$-th column of $T$ can be non-zero. Furthermore, the constraint $T\mathbf{1} = h_{\hat\theta}(\cdot|x)$ ensures that the $\kappa$-th column of $T$ actually equals $h_{\hat\theta}(\cdot|x)$. In other words, the set $\Pi(h_{\hat\theta(\cdot|x)}, \mathbb{e}_\kappa)$ contains only one feasible transport plan, so (3) can be computed directly as

$$W_p^p(h_{\hat\theta}(\cdot|x), \mathbb{e}_\kappa) = \sum_{\kappa' \in \mathcal{K}} M_{\kappa',\kappa} h_{\hat\theta}(\kappa'|x) = \sum_{\kappa' \in \mathcal{K}} d_{\mathcal{K}}^p(\kappa', \kappa) h_{\hat\theta}(\kappa'|x)$$

Now let $\hat\kappa = \operatorname{argmax}_\kappa h_{\hat\theta}(\kappa|x)$ be the prediction, we have

$$h_{\hat\theta}(\hat\kappa|x) = 1 - \sum_{\kappa \neq \hat\kappa} h_{\hat\theta}(\kappa|x) \geq 1 - \sum_{\kappa \neq \hat\kappa} h_{\hat\theta}(\hat\kappa|x) = 1 - (K-1)h_{\hat\theta}(\hat\kappa|x)$$

Therefore, $h_{\hat\theta}(\hat\kappa|x) \geq 1/K$, so

$$W_p^p(h_{\hat\theta}(\cdot|x), \mathbb{e}_\kappa) \geq d_{\mathcal{K}}^p(\hat\kappa, \kappa) h_{\hat\theta}(\hat\kappa|x) \geq d_{\mathcal{K}}^p(\hat\kappa, \kappa)/K$$

The conclusion follows by applying Theorem 5.1 with $p = 1$. ☐

## D Algorithmic Details of Learning with a Wasserstein Loss

In Section 5, we describe the statistical generalization properties of learning with a Wasserstein loss function via empirical risk minimization on a general space of classifiers $\mathcal{H}$. In all the empirical studies presented in the paper, we use the space of linear logistic regression classifiers, defined by

$$\mathcal{H} = \left\{ h_\theta(x) = \left( \frac{\exp(\theta_k^\top x)}{\sum_{j=1}^K \exp(\theta_j^\top x)} \right)_{k=1}^K : \theta_k \in \mathbb{R}^D, k = 1, ..., K \right\}$$

We use stochastic gradient descent with a mini-batch size of 100 samples to optimize the empirical risk, with a standard regularizer $0.0005 \sum_{k=1}^K \|\theta_k\|_2^2$ on the weights. The algorithm is described in Algorithm 2, where WASSERSTEIN is a sub-routine that computes the Wasserstein loss and its subgradient via the dual solution as described in Algorithm 1. We always run the gradient descent for a fixed number of 100,000 iterations for training.

---
**Algorithm 2** SGD Learning of Linear Logistic Model with Wasserstein Loss

---
Init $\theta^1$ randomly.
**for** $t = 1, \ldots, T$ **do**
    Sample mini-batch $\mathcal{D}^t = (x_1, y_1), \ldots, (x_n, y_n)$ from the training set.
    Compute Wasserstein subgradient $\partial W_p^p/\partial h_\theta|_{\theta^t} \leftarrow$ WASSERSTEIN$(\mathcal{D}^t, h_{\theta^t}(\cdot))$.
    Compute parameter subgradient $\partial W_p^p/\partial\theta|_{\theta^t} = (\partial h_\theta/\partial\theta)(\partial W_p^p/\partial h_\theta)|_{\theta^t}$
    Update parameter $\theta^{t+1} \leftarrow \theta^t - \eta_t \partial W_p^p/\partial\theta|_{\theta^t}$
**end for**

---

Note that the same training algorithm can easily be extended from training a linear logistic regression model to a multi-layer neural network model, by cascading the chain-rule in the subgradient computation.

## E Empirical study

### E.1 Noisy label example

We simulate the phenomenon of label noise arising from confusion of semantically similar classes as follows. Consider a multiclass classification problem, in which the labels correspond to the vertices on a $D \times D$ lattice on the 2D plane. The Euclidean distance in $\mathbb{R}^2$ is used to measure the

(a) Noise level 0.1                           (b) Noise level 0.5

Figure 8: Illustration of training samples on a 3x3 lattice with different noise levels.

semantic similarity between labels. The observations for each category are samples from an isotropic Gaussian distribution centered at the corresponding vertex. Given a noise level $t$, we choose with probability $t$ to flip the label for each training sample to one of the neighboring categories[8], chosen uniformly at random. Figure 8 shows the training set for a $3 \times 3$ lattice with noise levels $t = 0.1$ and $t = 0.5$, respectively.

Figure 2 is generated as follows. We repeat 10 times for noise levels $t = 0.1, 0.2, \ldots, 0.9$ and $D = 3, 4, \ldots, 7$. We train a multiclass linear logistic regression classifier (as described in section D of the Appendix) using either the standard KL-divergence loss[9] or the proposed Wasserstein loss[10]. The performance is measured by the mean Euclidean distance in the plane between the predicted class and the true class, on the test set. Figure 2 compares the performance of the two loss functions.

### E.2   Full figure for the MNIST example

The full version of Figure 4 from Section 6.1 is shown in Figure 9.

### E.3   Details of the Flickr tag prediction experiment

From the tags in the Yahoo Flickr Creative Commons dataset, we filtered out those not occurring in the WordNet[11] database, as well those whose dominant lexical category was "noun.location" or "noun.time." We also filtered out by hand nouns referring to geographical location or nationality, proper nouns, numbers, photography-specific vocabulary, and several words not generally descriptive of visual content (such as "annual" and "demo"). From the remainder, the 1000 most frequently occurring tags were used.

We list some of the 1000 selected tags here. The 50 most frequently occurring tags: *travel, square, wedding, art, flower, music, nature, party, beach, family, people, food, tree, summer, water, concert, winter, sky, snow, street, portrait, architecture, car, live, trip, friend, cat, sign, garden, mountain, bird, sport, light, museum, animal, rock, show, spring, dog, film, blue, green, road, girl, event, red,*

(a) Posterior prediction for images of digit 0.

(b) Posterior prediction for images of digit 4.

Figure 9: Each curve is the predicted probability for a target digit from models trained with different $p$ values for the ground metric.

*fun, building, new, cloud.* ... and the 50 least frequent tags: *arboretum, chick, sightseeing, vineyard, animalia, burlesque, key, flat, whale, swiss, giraffe, floor, peak, contemporary, scooter, society, actor, tomb, fabric, gala, coral, sleeping, lizard, performer, album, body, crew, bathroom, bed, cricket, piano, base, poetry, master, renovation, step, ghost, freight, champion, cartoon, jumping, crochet, gaming, shooting, animation, carving, rocket, infant, drift, hope.*

The complete features and labels can also be downloaded from the project website[12]. We train a multiclass linear logistic regression model with a linear combination of the Wasserstein loss and the KL divergence-based loss. The Wasserstein loss between the prediction and the normalized groundtruth is computed as described in Algorithm 1, using 10 iterations of the Sinkhorn-Knopp algorithm. Based on inspection of the ground metric matrix, we use $p$-norm with $p = 13$, and set $\lambda = 50$. This ensures that the matrix $\mathbf{K}$ is reasonably sparse, enforcing semantic smoothness only in each local neighborhood. Stochastic gradient descent with a mini-batch size of 100, and momentum 0.7 is run for 100,000 iterations to optimize the objective function on the training set. The baseline is trained under the same setting, using only the KL loss function.

To create the dataset with reduced redundancy, for each image in the training set, we compute the pairwise semantic distance for the groundtruth tags, and cluster them into "equivalent" tag-sets with a threshold of semantic distance 1.3. Within each tag-set, one random tag is selected.

Figure 10 shows more test images and predictions randomly picked from the test set.

(a) **Flickr user tags**: zoo, run, mark; **our proposals**: running, summer, fun; **baseline proposals**: running, country, lake.

(b) **Flickr user tags**: travel, architecture, tourism; **our proposals**: sky, roof, building; **baseline proposals**: art, sky, beach.

(c) **Flickr user tags**: spring, race, training; **our proposals**: road, bike, trail; **baseline proposals**: dog, surf, bike.

(d) **Flickr user tags**: family, trip, house; **our proposals**: family, girl, green; **baseline proposals**: woman, tree, family.

(e) **Flickr user tags**: education, weather, cow, agriculture; **our proposals**: girl, people, animal, play; **baseline proposals**: concert, statue, pretty, girl.

(f) **Flickr user tags**: garden, table, gardening; **our proposals**: garden, spring, plant; **baseline proposals**: garden, decoration, plant.

(g) **Flickr user tags**: nature, bird, rescue; **our proposals**: bird, nature, wildlife; **baseline proposals**: ature, bird, baby.

Figure 10: Examples of images in the Flickr dataset. We show the groundtruth tags and as well as tags proposed by our algorithm and baseline.

## Footnotes

[8]Connected vertices on the lattice are considered neighbors, and the Euclidean distance between neighbors is set to 1.

[9]This corresponds to maximum likelihood estimation of the logistic regression model.

[10]In this special case, this corresponds to weighted maximum likelihood estimation, c.f. Section C.

[11]http://wordnet.princeton.edu

[12] http://cbcl.mit.edu/wasserstein/