[Reviews · NeurIPS 2015]

Submitted by Assigned_Reviewer_1

In this paper, the authors propose to use the wasserstein loss for data fitting in multilabel/multiclass learning. This loss can encode specific losses for between classes error and will promote meaningfull errors (such as error between dog races) instead of costly errors

( for instance the recent Google Gorilla/people).

The authors discuss the use of a regularized version of wasserstein distance (mainly for computational reasons).They propose 2 potential extensions that can handle the difference of mass between the data and the predicted values. Short numerical experiments show the interest of the approach.

It is an good paper that propose a novel data fitting term that can encode error cost between classes. Numerical experiments seem encouraging. The rebuttal of the authors clarified a lot of my questions but there are problems that must be addressed before acceptance.

-The use of Wasserstein to encode between class loss is elegant but one can wonder if it's not a bit optimistic. Indeed the loss correspond to

the

affectation between class that minimize the loss. Intuitively, one would like to minimize the maximum loss over the residue |h()-min(h,y)| in order to avoid the worst case scenario (gorilla/human discussed above). Note that using the maximum loss on the residue would also boil down to TV for multiclass classification and 0-ground metric. This should be at least discussed in the introduction (with a figure?) since Wasserstein is a complex tool that is difficult to interpret.

- strategy 4.2.1 seem to just be lost space. If it has been tested and is inferior to the 4.2.2, then the authors should just refer to it in one or two sentences and spend more time detailing the model and the optimization problem.

- Add the demonstration for proposition 4.1 and 4.2 in the supplementary material. A discussion about the convergence of the fixed point iteration is also required since it will not provably converge unless the operation is a contraction mapping.

- Thanks to the rebuttal, we now know what the model of h(x) is (linear+softmax). This should appear clearly in the paper along with an equation giving the final optimization problem. This must be followed by a quick discussion on the chosen optimization algorithm and a discussion about the type of problem (is it convex? I doubt it). Not only

is it necessary for reproducible research but it will clarify the proposal.

- While you will doubtfully have time to add it in the final version, numerical experiments would benefice from comparison to other multilabel approaches (binary relevance for instance). One way to illustrate the strength of the proposed approach is to use a performance measure independent to the wasserstein (unlike K-cost that will obviously be better for wasserstein minimisation). For instance flicker you could use a clustering of the tags and you could measure the error using inter-cluster errors. If the clustering encode semantically similar tags together, your proposed approach will clearly perform better that other divergence loss (l1,l2,KL) since it will promote the semantic relations encoded in the metric.

Summary: Good paper leveraging the strengths of optimal transport for error-cost-aware

multilabel/multiclass learning. Some important discussions are missing and numerical experiments are a bit short but the paper should be accepted since the approach is clearly novel and might change the way classifiers are estimated.

Submitted by Assigned_Reviewer_2

Quality: The paper quality is high and includes important theoretical advances on learning with Wasserstein-based loss functions.

Related work:

Wasserstein distances have been applied in a wide variety of fields. It might be of interest to point out a few more instances, e.g. the use of Wasserstein distances in Topological Data Analysis. Background reference: Persistent Homology: Theory and Practice, H. Edelsbrunner, D. Morozov.

128-136: please provide a standard reference for this approach.

191-192: "(5) can be used directly as a surrogate for (2)" Explain to what extent this

provides a good approximation. Theorems?

Clarity: The writing clarity of the paper is excellent.

Originality: The paper appears to be highly original.

Significance: The theoretical contributions of this paper are likely to be significant, but it is difficult to place the experimental results relative to competing approaches. The paper would have benefited from a more thorough experimental comparison to other approaches.
Summary: The paper proposes a novel approach to learning based on Wasserstein loss by smoothing and approximating the exact Wasserstein loss that is computationally intractable for large datasets.

The authors present several theoretical contributions which appear to be significant, including a relaxation of the smoothed transport problem and theorems on statistical properties of the approach. Furthermore an experimental evaluation on Flickr images and the MNIST dataset is provided.

Submitted by Assigned_Reviewer_3

-- Convergence of the algorithm --

My major concern is that there is no claim of convergence of the algorithm. The authors do not even bother mentioning that they do not have any guarantee for their algorithm (they somehow put this under the carpet). My general feeling is that although deriving iterations as fixed point on the first order conditions is a possible heuristic, it is bad practice, in particular because it is quite hard to prove by hand contraction properties for the mapping. A better and easier practice is to use existing algorithms that are known to converge.

In this very specific case, I highly recommend the authors to use the so-called Dykstra algorithm (which is mentioned by the authors in Section 4.2.1), which, in the specific case of the entropic optimal transport, boils down to iterative Bregman projections, which in turn is equivalent to Sinkhorn. This is quite well documented in the paper [Benamou et al, SIAM CISC 2015]. When the marginal hard constraints are replaced by KL penalties, it is quite likely (although I did not bother checking this) that Dykstra's boils down to the iterations proposed by the authors. It might also lead to a slightly different algorithm, but hopefully with a similar Sinkhorn scaling structure. The rationale is that Dykstra is equivalent to a block alternate minimization on the two scaling variables on a dual problem. And also, the proximal operator (for the KL metric) of the function T->KL(T1|a) is easy to find and corresponds to a diagonal scaling, as can be seen by writing the first order condition. And this algorithm comes with convergence guarantees.

-- Numerical results --

Another concern is that the numerical part lacks a lot of details concerning the exact setup used, in particular: - what classification method is used (i.e. h_theta)? - how is the minimization performed with respect to theta ?

I see that some more details is given in the rebuttal, and I encourage the authors to further edit the final version of the paper in this direction.

-- Theoretical analysis --

I liked the fact that the author proposed some theoretical analysis in Section 5. However, it is quite unclear what conclusion should be drawn from the upper bound (9). In particular, how does this compare to other losses ?

Summary: I found this paper to be quite interesting: it presents an innovative statistical estimation procedure, and show promising numerical results. I especially liked the idea of using a KL penalty to handle un-normalized densities, which allows the authors to preserve the iterative scaling structure of Sinkhorn iterations. There are a few shortcoming, that I list bellow, that I think should be quite easy to fix. Despite these issues, I think this paper has the possibility to open new area of research, and thus I recommend acceptation.

Submitted by Assigned_Reviewer_4

This paper proposes to use the optimal transport distance, which the authors call the Wasserstein loss, as a loss function in learning. Theoretical analysis and some experiments are provided.

This paper does not have a coherent storyline. It is not clear what the authors are trying to do and what their main contributions are. The authors should try to separate their contributions and results that are already available in the literature.

The learning problem, as stated by the authors, is in Section 3.1. I assume this is (one of) the main goal of the paper. With the Wasserstein loss function, how does one solve this problem? Under which hypothesis space? What is the algorithm? Does it

always have a solution? Is the solution unique? What is the computational complexity? None of these questions have been answered.

Given that the distance function (2) is well-known and the regularization (5) is from Cuturi [16], what are your main contributions here? They need to be clearly stated. Note that this entire section is concerned with the computation of the loss function (2), *not* in solving the above learning problem.

Propositions 4.1 and 4.2: I don't see the proofs for these propositions.

Formula (2): What is the inner product here? Is it a Frobenius inner product?

Experiments:

The authors should test their framework much more extensively on multi-label learning problems. The current experiments seem quite limited. It is not at all clear what is being implemented here either. Are you solving the learning problem stated in Section 3.1? By which algorithm?

Figure 3: What if alpha = 0, so that there is only the Wasserstein loss component?
Summary: This paper contains some interesting ideas, but they need to be worked out much more

extensively for the paper

to become publishable.

Submitted by Assigned_Reviewer_5

This paper motivates the use for Wasserstein loss for learning algorithms. The paper presents the Wasserstein loss as a robust loss and provides some good results, mainly:- - The idea of using Wasserstein loss, - Probabilistic bound for the expected Wasserstein loss, - Connection with other losses,

While the paper is interesting, I find some major concerns:

1. The paper has a number of incomplete/inconsistent descriptions. For example,

a. The optimization formulation in eq. (8) needs to be properly explained (What are a,b,gamma_a,gamma_b etc.) Further, based on Proposition 4.1 it seems like the constraint for T* depends on T* itself. How is this proposition utilized to solve eq. (8)?

b. What is the advantage of eq.(8) over eq (4)? How is the formulation (8) solved? What is the time complexity? The paper doesn't seem to justify/discuss these.

c. The paper proposes to be use ERM. However, the loss function defined in eq (8) seems to be using a relaxation term which in turn seems to control the structure of the concept classes. This is in line with most robust loss functions like, hinge loss (with functional margin), epsilon loss (with insensitive zone) which partially controls the structure of the concept classes and is a part of the SRM approach. The paper needs more discussion on eq (8) and how it still follows the ERM considering the lamda, gamma_a, gamma_b parameters.

2. The Structure of the paper needs improvement: a. Avoid experimental results in the Introduction section if not providing enough context/info to be able to understand them

3. The paper needs to improve on experiments/results/inferences: a. Section 6.1 : The experiment lacks a brief description of the data used. Note there are multiple versions of MNIST data with different partition of train/test data. b. Section 6.2 How is the hypothesis/concept class parameterized? Is it linear/non-linear?

c. Section 6.2 : What is the ground metric used? In the definition of performance measure what is d_k used to compute the top K-cost? Further, why us the top-K cost used as a measure? Typically most experiments regarding the datasets use accuracy.

d. Section 6.2: A brief explanation of the data used i.e. Train/test partitioning/ experimental setting is missing. Further a good description of the problem i.e. Is this a multi-class problem? What are the class prior probabilities? etc are also missing. e. How were the model parameters in eq. (8) i.e. lamda, gamma_a,gamma_b selected in the experimental results?

In all the paper needs major improvement on the above mentioned points.

Summary: Interesting paper with well motivated problem. However, need significant improvement in detailing the approach, experiments and justifications.

Author Feedback
Author rebuttal: Thanks for the valuable feedback from all reviewers! Due to limited space we address only the main concerns here. We will add all clarifications and missing references to the final version.

### R1

Q2: We agree that partial transport is not a strong approach. It universally underperformed soft KL relaxation in our experiments and was included mainly to motivate soft KL. We will clarify this in the final version. Thanks for the idea of using a "hole" class to absorb extra mass. We hadn't investigated this and it's very interesting.

Q3,Q4: [Details of experiments] We solved ERM (Sec. 3) via stochastic gradient descent, using the methods in Sec. 4 to compute gradients. Specifically, we used normalized target labels and Sinkhorn iterations for the gradient, primarily because the KL-divergence (with which we combine in Sec. 6.2) requires normalized outputs. Soft KL performed similarly, showing the same performance effect as in Fig. 3 and 4. The hypothesis space consists of linear logistic regression models (i.e. linear+softmax). Hyperparameters were chosen by cross validation.

Q5: Fig. 4 shows performance vs. alpha, including alpha=0. The values of the two losses might be hard to compare directly, but we should be able to compare the norms of their parameter gradients. Thanks for the suggestion.

### R3

Q1a: [Notation in (8)] We inadvertently changed notation in equation (8) and Prop. 4.1 and 4.2. alpha and beta should be u and v (as in previous sections), while a and b should be h(x) and y, respectively. Thanks for pointing this out.

Q1a: [Prop. 4.1 states T^* recursively] Prop. 4.1 intends only to show that an optimal transport matrix for (8) must equal the matrix K left- and right-multiplied by diagonal matrices. These matrices, as noted by the reviewer, are stated recursively, in terms of the optimal transport matrix. This form is used to derive the fixed point iteration given in Prop. 4.2. We will add proofs for 4.1 and 4.2 to the supplement.

Q1b: [Advantage of (8) over (4)] Note that Sec. 4 focuses on an efficiently evaluated surrogate for (4), given by equation (5). (8) extends (5) from normalized to unnormalized measures; this is the intent of Sec. 4.2.

Q1b: [Solving (8)] We intended that the concrete algorithm for solving (8) would be suggested by Prop. 4.2. It is exactly the Sinkhorn-Knopp algorithm for (5), except that the assignments for u and v (here alpha and beta) are replaced by those given in Prop. 4.2. We will clarify this in the final version.

Q1b: [Time complexity] The complexity for solving (5) is exactly that of the Sinkhorn-Knopp algorithm and was documented in [16]. The complexity of each iteration for (8) is the same as that for (5) -- it is dominated by two matrix-vector products, given in Prop. 4.2. As was shown for (5) in [16], the number of iterations to converge for (8) is nearly independent of problem size. We will clarify this in the final version.

Q1c: [Relation of entropy regularizer to Structural Risk Minimization] Both the entropy regularizer and the KL penalties apply to the transport matrix, rather than the learning parameters. It would be interesting, though, to see how the combination with SRM affects the statistical bounds. Thanks for suggesting this.

Q3a: We used the standard MNIST dataset from Y. LeCun's webpage.

Q3c: We reported both top-K cost and accuracy (AUC). Please see Fig. 4, and also discussion in lines 368-371. We used top-K cost to demonstrate that our loss leads to predictions semantically closer to ground truth.

Q3d: The ground metric d_K is the Euclidean distance between word2vec embeddings. The Flickr experiment is a 1000-way multi-label problem. Please see lines 322-323 and 353-355 as well as (R1 Q3) above for details of the experiments.

### R4

Our experiments try to demonstrate that incorporating a natural metric on outputs into the loss can both smooth predicted outputs (Sec. 6.1) and move them closer to the ground truth (Sec. 6.2). Our baseline is therefore a widely-used standard loss that doesn't use the output metric.

### R6

[Storyline and contributions] We develop a loss for multilabel learning given a natural metric on the output space. It is, to our knowledge, the first use of the Wasserstein distance as a loss for supervised learning. We present algorithms for efficiently computing the gradient of the loss, including a novel extension of the Wasserstein distance to unnormalized measures, theoretical motivation for its use via a novel statistical learning bound and relation to the Jaccard index, and empirical demonstrations of both a smoothing effect on the predicted outputs and improved performance on a real-data tag prediction task. We will try to clarify this in the final version.

Note that Sec. 4 concerns computing not the loss but rather its gradient, which is needed to solve the learning problem by gradient descent.

[Fig. 3 and 4 when alpha=0] Please see (R1 Q5).